# MERT: Acoustic Music Understanding Model with Large-Scale Self-supervised Training

**Yizhi Li** [1,2][*]  **Ruibin Yuan** [3,4][*]  **Ge Zhang** [4,5,6][*]  **Yinghao Ma** [7][*]  **Xingran Chen**  **Hanzhi Yin** [3]  **Chenghao Xiao** [8]

**Chenghua Lin** [1,2][†]  **Anton Ragni** [2]  **Emmanouil Benetos** [7]  **Norbert Gyenge** [2]  **Roger Dannenberg** [3]  **Ruibo Liu** [9]

**Wenhu Chen** [5]  **Gus Xia** [10,11]  **Yemin Shi** [6,12]  **Wenhao Huang** [6]  **Zili Wang**  **Yike Guo** [4]  **Jie Fu** [4,6][†]

m-a-p.ai  [1]University of Manchester  [2]University of Sheffield  [3]Carnegie Mellon University

[4]Hong Kong University of Science and Technology  [5]University of Waterloo  [6]Beijing Academy of Artificial Intelligence

[7]Queen Mary University of London  [8]Durham University  [9]Dartmouth College  [10]MBZUAI  [11]New York University  [12]linksoul.ai

## Abstract

Self-supervised learning (SSL) has recently emerged as a promising paradigm for training generalisable models on large-scale data in the fields of vision, text, and speech. Although SSL has been proven effective in speech and audio, its application to music audio has yet to be thoroughly explored. This is partially due to the distinctive challenges associated with modelling musical knowledge, particularly tonal and pitched characteristics of music. To address this research gap, we propose an acoustic **M**usic und**ER**standing model with large-scale self-supervised **T**raining (**MERT**), which incorporates teacher models to provide pseudo labels in the masked language modelling (MLM) style acoustic pre-training. In our exploration, we identified an effective combination of teacher models, which outperforms conventional speech and audio approaches in terms of performance. This combination includes an acoustic teacher based on Residual Vector Quantisation - Variational AutoEncoder (RVQ-VAE) and a musical teacher based on the Constant-Q Transform (CQT). Furthermore, we explore a wide range of settings to overcome the instability in acoustic language model pre-training, which allows our designed paradigm to scale from 95M to 330M parameters. Experimental results indicate that our model can generalise and perform well on 14 music understanding tasks and attain state-of-the-art (SOTA) overall scores.

## 1 Introduction

Pre-trained language models (PLMs) can learn generalisable representations of data without human annotated labels in a self-supervised learning (SSL) style, leading to remarkable performance improvement in natural language processing and related fields (Brown et al., 2020; Fang et al., 2022; Chen et al., 2021a). Music is widely recognised as a special language that can be used to communicate across different cultures (Mehr et al., 2019). The internal similarity between music and language as a communication interface lays a promising foundation for adapting PLM-based methods to model music sequences. We argue that the benefit is twofold. First, PLMs can potentially pave the way to *unify* the modelling of a wide range of music understanding, or the so-called Music Information Retrieval (MIR) tasks, including but not limited to music tagging, beat tracking, music transcription, source separation, etc., so that different tasks no longer need task-specific models or features. Second, releasing a PLM for acoustic music understanding allows the redistribution of the musical knowledge rather than the data itself, which saves the costs of manual annotation and copyright law restrictions.

Unfortunately, we are yet to see a general-purpose and cost-effective open-source PLM on *acoustic* music understanding. Most existing studies are designed to solely address music tagging problems (Pons and Serra, 2019; Spijkervet and Burgoyne, 2021; McCallum et al., 2022; Huang et al., 2022; Zhu et al., 2021; Zhao and Guo, 2021), and many of them do not provide open-source code bases or checkpoints for further evaluation. A promising model is JukeMIR (Castellon et al., 2021), which is based on Jukebox (Dhariwal et al., 2020) and provides a comprehensive evaluation on MIR

---

[*]The authors contributed equally to this work.

[†]Corresponding authors.

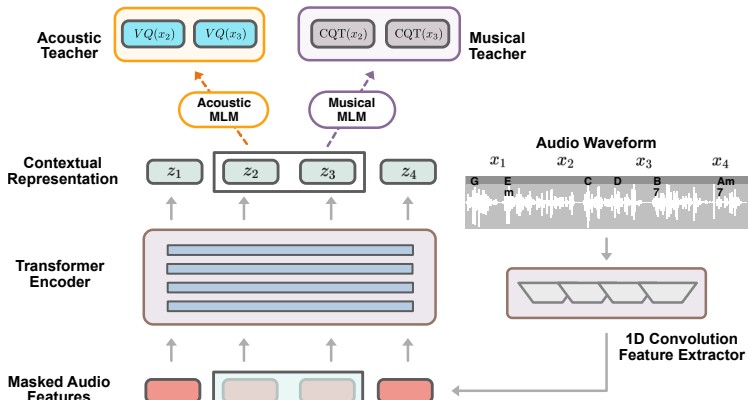

Figure 1: Illustration of the MERT Pre-training Framework.

downstream tasks. However, this foundation model uses cumbersome hierarchical auto-regressive transformer decoders containing billions of parameters to model music audio, leading to significant inefficiency for conducting general music understanding tasks (e.g., it takes weeks to inference on datasets like MTG (Bogdanov et al., 2019) with a consumer-grade 3090 GPU).

The aforementioned research gap has urged us to design and open-source a *generalisable* and *computationally affordable* pre-trained acoustic music model. In this paper, we propose an acoustic **M**usic und**ER**standing model with large-scale self-supervised **T**raining (**MERT**). MERT inherits a speech SSL paradigm, employing teacher models to generate pseudo targets for sequential audio clips. Specifically, to capture the distinctive pitched and tonal characteristics in music, MERT incorporates a multi-task paradigm to balance the *acoustic* and *musical* representation learning as demonstrated in Fig. 1. In the proposed design, an Residual Vector Quantisation - Variational Autoencoder (RVQ-VAE) (Défossez et al., 2022) is used as the *acoustic teacher* to provide discretised acoustic-level summarisation of the music signal. The Constant-Q Transformation (CQT) (Brown, 1991) model is further introduced as the *music teacher* for capturing pitch and harmonic inductive bias. Regarding the context dependencies and music hierarchies, as indicated in Borsos et al. (2022), we leave the task of modelling high-level and abstract patterns to the stacked layers of self-attentions in the transformer.

We also explore a wide range of settings for the transformer and 1D convolution encoder to overcome the instability in acoustic model pre-training, which permits effective scaling up of MERT from 95M to 330M size when blending acoustic and musical knowledge. By scaling up to 330M size (only 7% the size of JukeBox), MERT achieves overall state-of-the-art (SOTA) results on various MIR tasks, which demonstrates a strong generalisability on music understanding. Last but not least, we analyse multiple pre-trained settings considering the teachers and share our decision routes § 5.2 and § 5.3, which could potentially guide future acoustic music understanding pre-training research.

To summarise, our contributions are:

- proposing a multi-task style predictive acoustic self-supervised learning paradigm, which achieves SOTA performance on various MIR tasks, including important yet unexplored tasks for pre-training such as pitch detection, beat tracking and source separation applications;
- conducting a broad range of analysis based on ablation study of the proposed MERT pre-training paradigm;
- exploring robust and stable strategies for acoustic music model training to overcome training instability and frequent crashes when scaling up the pre-training on model size;
- providing an open-source, generalisable and computationally affordable acoustic music pre-trained model, which addresses the needs of both industry and research communities.

## 2 RELATED WORK

**PLMs for Acoustic Music** The field of music information retrieval (MIR) has long been facing challenges in data availability due to the costs associated with music audio annotation and country-specific copyright laws (Chen et al., 2019; Castellon et al., 2021). To address this challenge, pre-trained language models (PLMs) for acoustic music have been proposed to provide reusable learned

representations, enabling transfer learning for various downstream MIR tasks without the need for extensive data annotation (Castellon et al., 2021). However, current acoustic music pre-trained models still have room for improvement in terms of providing open-source, generalisable, and lightweight learned representations suitable for both industrial and research applications (McCallum et al., 2022). Existing acoustic music pre-trained models primarily focus on tagging tasks and rely on supervised tagging labels for pre-training (Pons and Serra, 2019; Spijkervet and Burgoyne, 2021; McCallum et al., 2022; Huang et al., 2022). While some studies have explored contrastive learning for acoustic music pre-training, they face limitations in training data and model size, hampering the performance improvements (Choi et al., 2017; Li et al., 2022). Additionally, several models trained on inaccessible datasets or without publicly available codes and model weights make it difficult to reproduce or extend their approaches (McCallum et al., 2022; Castellon et al., 2021; Li et al., 2022; Zhu et al., 2021; Zhao and Guo, 2021). Although some general-purpose audio representation models show potential for music audio representation learning, their performance is mostly evaluated on limited MIR downstream tasks (Saeed et al., 2021; Borsos et al., 2022; Wang et al., 2023). This lack of comprehensive evaluation hinders further studies and a thorough understanding of the pros and cons of existing models.

**Self-Supervised Speech Processing** Music and speech processing are closely related (Jasmin et al., 2020) since they usually use the same audio data formats. Additionally, both acoustic music and speech processing models need to deal with the cocktail party problem (Brown and Bidelman, 2022; Petermann et al., 2022) since good source separation capabilities help both separating noises and background sounds with speech and processing polyphonic music audio. These common grounds between music and speech processing inspire us to adapt SOTA speech pre-trained models and tailor them specifically for music audio processing tasks. For instance, existing research work targeting general-purpose audio representations (Saeed et al., 2021; Borsos et al., 2022; Wang et al., 2023) has verified that self-supervised speech processing models can be extended beyond speech to downstream entry-level music tasks, including generating mono piano music and music reconstruction.

**Audio Representation with Language Modelling** Mask strategy-based large-scale language models have been applied to a wide range of domains (Lample and Charton, 2019; Chen et al., 2021a;b; Fang et al., 2022), but still remain under-explored in acoustic music understanding. For audio, Dhariwal et al. (2020) investigates generating hierarchical tokens which can be further employed to reconstruct music, inspiring subsequent research to understand and generate acoustic music based on extracted discrete tokens from continuous features. Baevski and Mohamed (2020) introduce a pre-trained VQ-VAE (Baevski et al., 2019) to provide prediction targets to conduct speech representation learning with MLM. While introducing K-means to provide discrete token codebooks and pre-training the model to detect sound units, Hsu et al. (2021) claim that a better teacher model in SSL could lead to better downstream task performance. Additionally, recent speech processing pre-trained models (Borsos et al., 2022; Wang et al., 2023) propose to train or adopt separately trained codecs (Zeghidour et al., 2021; Défossez et al., 2022) for discrete token extraction. Based on the conclusion from previous studies, the recently released RVQ-VAEs (Zeghidour et al., 2021; Défossez et al., 2022), achieving good results in music reconstruction, could be adopted as teacher models for music understanding pre-training and provide acoustic information guidance. Yet some of the uniqueness of music processing such as timbre and harmony remains unexplored. We thus propose to incorporate a corresponding musical teacher model in MERT to fill such an important gap.

## 3 METHODOLOGY

This section introduces the pre-training paradigm and architecture of our models. It includes prediction to acoustic teachers such as k-means or deep music features, and reconstruction to music teachers such as CQT spectrum, both based on the well-established masked language modelling (MLM) .

### 3.1 PRE-TRAINING WITH MLM

Supervised Learning requires a labelled dataset $\mathcal{D}_t = \{x_i^{(t)}, y_i^{(t)}\}_{i=1}^N$. Here, $N$ is the number of data samples, $x_i^{(t)}$ is the $i^{th}$ data sample in the dataset, and $y_i^{(t)}$ is the corresponding label. From $\mathcal{D}_t$, we can train a machine learning algorithm $f_\theta(\cdot)$ parameterised with $\theta$ that makes label predictions on

each data sample. Unsupervised learning, in contrast, learns an algorithm based on an unlabelled dataset $\mathcal{D} = \{x_i\}_{i=1}^M$, with SSL being a specific type of this class. For each data sample $x_i$, SSL derives a new data $x_i'$ with a pseudo label $y_i'$. The training process is to minimise the loss between each pseudo label $y_i'$ and the prediction based on new data $\hat{y}_i = f_\theta(x_i')$ as denoted in Eq.1.

$$\theta^* = arg\,min_\theta \sum_{x_i^{(t)} \in D} \mathcal{L}\left(f_\theta(x_i'^{(t)}), y_i'^{(t)}\right). \tag{1}$$

MLM is a famous example of pseudo-label generation. Let $x_i = \left[x_i^{(1)}, x_i^{(2)}, \cdots, x_i^{(L)}\right]$ be the $i^{th}$ data sample in a sequential dataset with length $L$, and $M \subset [L]$ is a subset of indices randomly chosen from 1 to $L$. Then, the new data is defined by the following equation

$$x_i' = \left[\mathbf{1}_{[L]\backslash M}(1) \cdot x_i^{(1)}, \mathbf{1}_{[L]\backslash M}(2) \cdot x_i^{(2)}, \cdots, \mathbf{1}_{[L]\backslash M}(L) \cdot x_i^{(L)}\right] \tag{2}$$

where $\mathbf{1}_{[L]\backslash M}(x)$ denotes the indicator function, that is, $\mathbf{1}_{[L]\backslash M}(x) = 1$ if and only if $x$ is outside the masked indices set $M$. The pseudo-label that needs to be learned is typically $y_i' = x_i - x_i'$, i.e., the masked data. However, reconstructing masked data $y'$ for raw audio tasks as pseudo-label is hard to train. HuBERT (Vaswani et al., 2017; Hsu et al., 2021) uses a dimension-reduced feature $z'$ derived from $y'$ with phonetic acoustic information, which forms the design basis of our pre-training strategy.

As a speech SSL system, HuBERT utilises offline clustering to acquire pseudo labels for a BERT-like prediction loss. Specifically, it uses Mel-frequency cepstral coefficients (MFCCs), a widely-used traditional feature in speech-related tasks, as acoustic features for clustering. The obtained results are then utilised as pseudo labels in the first iteration of pre-training. It then uses the learned representation for clustering to get a pseudo label for the second iteration pre-training. Such a pseudo label includes acoustic information in human speech and can be aligned to phonemes. The loss functions of HuBERT are formulated as follows:

$$\mathcal{L}_H(f; x, M, Z) = \sum_{t \in M} \log p_f(z_t \mid x', t) \tag{3}$$

where $\log p_f(\cdot \mid x', t)$ is the log-likelihood function on clustering results given the masked input $x'$ and position $t$ derived from $f$; likelihood function $p_f$ is the Noise Contrastive Estimation (NCE) loss which is defined as

$$p_f(c \mid x', t) = \frac{\exp(\text{sim}(T(o_t), e_c)/\tau)}{\sum_{c'=1}^C \exp(\text{sim}(T(o_t), e_{c'})/\tau)}, \tag{4}$$

Here, $c \in [C]$ is a codeword of the clustering results and $e_c$ represents its embedding; sim is the cosine similarity; $o_t$ is the output of the model at timestep $t$; and $T(o_t)$ is the linear transformation of $o_t$, making it have the same dimension as $e_c$. Besides, $\tau$ scales the logit which is set to 0.1 in HuBERT. The linear transformation $T$, the model to generate outputs, and the embedding of all the clustering results are all learnable.

Overall, we use the same model as HuBERT but introduce several notable variations tailored to music. Specifically, we designed a better hidden-unit $z$ as pseudo tags for pre-training with multiple music acoustic features. In addition, we added a reconstruction loss to music features and employed additional music augmentation tricks.

### 3.2 MODELLING ACOUSTIC INFORMATION

The MFCC features are only good at modelling acoustic and timbre information for single-pitch signals, and therefore, the clustering results do not provide much timbre information in music recording. We proposed two potential approaches as the teacher on acoustic information: one based on traditional features, and the other based on deep learning.

The first method uses k-means on the log-Mel spectrum and Chroma features for timbre and harmonic acoustic information, respectively. In the case of music representation, each frame contains more information compared to speech, necessitating a larger number of classes for k-means clustering. The complexity of the k-means algorithm is linear with the number of centroids (clustering centres), leading to a time-consuming k-means for the music feature. To tackle this problem, we employ

300-means for the log-Mel spectrum with dimension 229, and 200-means for Chroma features with dimension 264, resulting in a total of 60,000 classes (200 centroids for Chroma features multiplied by 300 centroids for the log-Mel spectrum). Despite the increased number of classes, the computational complexity remains comparable to that of HuBERT. The disadvantage of k-means is that it is difficult to scale up to a larger number of classes and larger datasets, and the results are sensitive to initialisation.

The second choice for our acoustic teacher is EnCodec (Défossez et al., 2022), a recent learnable feature with 8-layer residual Vector Quantized-Variational AutoEncoder (VQ-VAE). Each acoustic feature, denoted as $z_{enc} \in [C]^{L \times 8}$, is a 2-dimensional auditory code matrix, and $L$ is the length of the recording. The row vector of each matrix $z_{enc}[t, :]$ represents the results of 8 different clusterings for frame $t$, and the column vector of each matrix $z_{enc}[:, j]$ represents the results from the $j^{th}$ codebook of the audio sequence, where $j \in \{1, \ldots, 8\}$. EnCodec converts 24kHz input waveforms to 8 different embeddings at 75Hz with a 320-fold reduction, and the quantizer has 1024 dimensions. In this setting, for each 5-second waveform, the discrete acoustic feature is a matrix with $375 \times 8$ entries, representing 375 frames (75Hz × 5s) and 8 deep acoustic features. With these embeddings, the decoder of EnCodec can reconstruct the waveform at 24 kHz with authentic information in timbre.

### 3.3 Modelling Musical Information

Apart from acoustic information, we added a new reconstruction loss to the Constant-Q transform (CQT) spectrogram to emphasise pitch-level information. The CQT is a type of frequency transform that is widely used in various MIR tasks, such as pitch detection, chord recognition, and music transcription. It is similar to the Fourier transform, but bin widths are proportional to frequency rather than equal, giving each octave the same number of bins, resulting in a better time-frequency trade-off for music audio where multiple pitches occur in multiple octaves. We utilize mean squared error (MSE) loss to reconstruct the CQT spectrum $z_{cqt}$ from the masked input audio $x'$. That is,

$$\mathcal{L}_{CQT}(f_{cqt}; x, M, \mathbf{z}_{cqt}) = \sum_{t \in M} \|z_{cqt,t} - f_{cqt}(x')_t\|_2 \tag{5}$$

And the final loss function $\mathcal{L}$ is a linear combination of both the acoustic loss function $\mathcal{L}_H$ and the musical-pitch loss function $\mathcal{L}_{CQT}$:

$$\mathcal{L} = \alpha \cdot \mathcal{L}_H + \mathcal{L}_{CQT} \tag{6}$$

## 4 Experiments

### 4.1 Evaluation Protocol

**Downstream Tasks** We evaluate our method and compare it with baseline models on 14 downstream tasks, including frame-level classification or regression tasks like music tagging, key detection, genre classification, emotion score regression, instrument classification, pitch classification, vocal technique detection, and singer identification; and sequential tasks like beat tracking and source separation. For instrument classification, we use the Nsynth (Engel et al., 2017) and MTG-instrument datasets, with receiver operating characteristic (ROC), and average precision (AP). The NSynth dataset is also used for pitch classification, with accuracy (ACC) as the evaluation metric. Vocal technique detection and singer identification based on the VocalSet dataset (Wilkins et al., 2018), with accuracy as the metric. For music tagging, we utilise the MagnaTagATune (MTT) (Law et al., 2009) and MTG-Jamendo (Bogdanov et al., 2019) datasets, averaging multiple embeddings for long audio recordings. Key detection is accomplished using the Giantsteps and Giantsteps-MTG-keys datasets (Knees et al., 2015; Korzeniowski and Widmer, 2017), with a refined accuracy (ACC$^{\text{refined}}$) metric. Genre classification is performed using the GTZAN (Tzanetakis and Cook, 2002) and MTG-Genre datasets, with ROC, and AP metrics. Emotion score regression is conducted on the Emomusic dataset (Soleymani et al., 2013), with the coefficient of determination (R2 score) of arousal and valence as evaluation metrics. Beat tracking is conducted on the GTZAN Rhythm dataset (Marchand and Peeters, 2015), using the F-measure (F1). Finally, source separation is accomplished using the MUSDB18 dataset (Rafii et al., 2017), with the Source-to-Distortion Ratio (SDR) as the evaluation metric. The full descriptions of the datasets and tasks can be found in Appendix B.1.

**Probing Protocol** Following Castellon et al. (2021); Yang et al. (2021), we restrict the testing protocol with probing rather than fine-tuning, i.e. freezing the backbone pre-trained models as deep feature extractor and only train a simple downstream structure, typically a multilayer perceptron (MLP) for frame-level tasks. For a fair comparison, we also limit the space for hyper-parameters searching. For full details please refer to Appendix B.2.

## 4.2 BASELINE METHODS

We select models pre-trained with various paradigms from both music and speech domains as our baselines to provide a comprehensive evaluation of the generalisation ability of the designs. MusiCNN (Pons and Serra, 2019) is selected as a representative supervised method, which is pre-trained with supervision from the Million Song Dataset tags (Bertin-Mahieux et al., 2011). CLMR (Spijkervet and Burgoyne, 2021) and MULE (McCallum et al., 2022) are selected as representatives of SOTA music representations trained with contrastive learning. Jukebox (Dhariwal et al., 2020) and the corresponding transfer learning method, JukeMIR (Castellon et al., 2021) is selected as the representative of transfer learning from a large-scale generative pre-trained musical representation. We also select the recently proposed strong speech SSL models, HuBERT (Hsu et al., 2021) and data2vec (Baevski et al., 2022), as our baselines since they share the same MLM pre-training paradigm with MERT. While HuBERT reconstructs the masked discrete tokens provided by the K-means teacher, data2vec uses the student model updated with an exponential moving average gradient to produce continuous representations for MLM prediction. In order to reveal the effectiveness of the pre-training paradigm itself rather than the training data distribution, we re-train the speech models and denote them by HuBERT$^{music}$ and data2vec$^{music}$. Additionally, we present the current SOTA for each task including results from both supervised and self-supervised methods.

## 4.3 IMPLEMENTATION DETAILS

**Training Settings** We deploy the proposed SSL architecture in the training of various model sizes with matched scales of data. We mined 160K hours of music recordings from the Internet to build a large-scale music dataset. Accordingly, the base size models (95M) are trained with a 1K hours subset whereas the whole dataset is used for the large model (330M). Specifically, we provide a special edition of the base model, `MERT-95M-public`, that is trained on a totally publicly available music dataset, music4all (Santana et al., 2020), with a data size of 910 hours. In the context of self-attention, the computational complexity scales quadratically with the sequence length. Therefore, when dealing with limited computational resources, there exists a trade-off between the batch size and the sequence length. In our preliminary experiments, we have observed that increasing the batch size provides greater performance improvements compared to extending the context length. To allow a larger batch size under the computational limitation, we adopt a strategy of randomly truncating audio clips into 5-second segments following Ma et al. (2023). This duration roughly corresponds to a 2-bar context in music. It is worth noting that our model utilises a convolutional relative positional embedding, similar to the approach introduced by Baevski et al. (2020) in Wav2Vec, enabling it to operate effectively in longer contexts, if required. The effective batch sizes and learning rates for the base model and large model are set to $1.5$ and $5.5$ hours, and their learning rates are set to $5e{-}4$, $1.5e{-}3$, respectively. Pre-training is carried out with the fairseq[1] framework. Models are trained with 64 A100-40GB GPUs with fp16. We also implement a data augmentation of randomly adding short segments to improve the representation robustness, and describe the details in Appendix A.1

**Training Stability** In our empirical findings, we observe that when scaling up acoustic encoder-only models, they tend to exhibit a higher susceptibility to training instability compared to models of similar size in text or image domains. Such instability can result in decreased performance or, in extreme cases, even lead to crashes in model training. During experimentation with scaling up to the `MERT-330M` model, we encounter notable instability manifested by constant gradient clipping and sporadic spikes in losses. This instability has a detrimental effect on the accuracy of MLM predictions and results in decreased performance on downstream tasks. Our attempts to resume training from previously saved checkpoints and data batches are proved unsuccessful in mitigating the instability issue. Furthermore, we observe that reducing the learning rate in this context not only fails to address the issue but also leads to a decline in performance and hindered the training convergence. We

---

[1]https://github.com/facebookresearch/fairseq

further explore the effectiveness of a seemingly-powerful method DeepNorm (Wang et al., 2022a) in stabilising acoustic language model pre-training, but find it to be ineffective. Eventually, we discover that incorporating attention relaxation techniques (Chen et al., 2021b) is beneficial in addressing the instability challenges. We also found that transitioning from post-layer normalisation (Post-LN) to pre-layer normalisation (Pre-LN) offers a potential solution of allowing training to continue. More information can be found in Appendix B.3.

Table 1: Experimental Performances of MERT and Baselines on Downstream Tasks (1/2). The baselines are grouped by supervised and unsupervised pre-training paradigms. The superscripts denote the category of the acoustic teacher used by MERT models. "public" refers to the MERT model trained with only open-source dataset. Results with star* are claimed in the references.

| Dataset
Task | MTT
Tagging | | GS
Key | GTZAN
Genre | GTZAN
Rhythm | EMO
Emotion | | Nsynth
Instrument | Pitch | VocalSet
Tech | VocalSet
Singer |
|---|---|---|---|---|---|---|---|---|---|---|---|
| Metrics | ROC | AP | Acc$^{Refined}$ | Acc | F1$^{beat}$ | R2$^V$ | R2$^A$ | Acc | Acc | Acc | Acc |
| MusiCNN [41] | 90.6* | 38.3* | 12.8* | 79.0* | - | 46.6* | 70.3* | 72.6 | 64.1 | 70.3 | 57.0 |
| CLMR [48] | 89.4* | 36.1* | 14.9* | 68.6* | - | 45.8* | 67.8* | 67.9 | 47.0 | 58.1 | 49.9 |
| Jukebox-5B [15; 57] | 91.5* | **41.4*** | 66.7* | 79.7* | - | **61.7*** | 72.1* | 70.4 | 91.6 | 76.7 | 82.6 |
| MULE [36] | 91.4* | 40.4* | 66.7* | 73.5* | - | 57.7* | 70.0* | 74.0* | 89.2* | 75.5 | **87.5** |
| HuBERT-base$^{music}$ [25] | 90.2 | 37.7 | 14.7 | 70.0 | **88.6** | 42.1 | 66.5 | 69.3 | 77.4 | 65.9 | 75.3 |
| data2vec-base$^{music}$ [2] | 90.0 | 36.2 | 50.6 | 74.1 | 68.2 | 52.1 | 71.0 | 69.4 | 93.1 | 71.1 | 81.4 |
| MERT-95M$^{K\text{-}means}$ | 90.6 | 38.4 | 65.0 | 78.6 | 88.3 | 52.9 | 69.9 | 71.3 | 92.3 | 74.6 | 77.2 |
| MERT-95M-public$^{K\text{-}means}$ | 90.7 | 38.4 | 67.3 | 72.8 | 88.1 | 59.7 | 72.5 | 70.4 | 92.3 | 75.6 | 78.0 |
| MERT-95M$^{RVQ\text{-}VAE}$ | 91.0 | 39.3 | 63.5 | 78.6 | 88.3 | 60.0 | **76.4** | 70.7 | 92.6 | 74.2 | 83.7 |
| MERT-330M$^{RVQ\text{-}VAE}$ | 91.3 | 40.2 | 65.6 | 79.3 | 87.9 | 61.2 | 74.7 | 72.6 | **94.4** | **76.9** | 87.1 |
| (Previous) SOTA | **92.0** [26] | **41.4** [15] | **74.3** [30] | **83.5** [36] | 80.6 [24] | **61.7** | 72.1 [15] | **78.2** [53] | 89.2 [36] | 65.6 [55] | 80.3 [39] |

Table 2: Experimental Performances of MERT and Baselines on Downstream Tasks (2/2). Average scores across *task* are calculated on the SOTA results and models applicable to all the tasks.

| Dataset
Task | MTG
Instrument | | MTG
MoodTheme | | MTG
Genre | | MTG
Top50 | | MUSDB
Source Seperation | | | | Avg. |
|---|---|---|---|---|---|---|---|---|---|---|---|---|---|
| Metrics | ROC | AP | ROC | AP | ROC | AP | ROC | AP | SDR$^{vocals}$ | SDR$^{drums}$ | SDR$^{bass}$ | SDR$^{other}$ | |
| MusiCNN [41] | 74.0 | 17.2 | 74.0 | 12.6 | 86.0 | 17.5 | 82.0 | 27.5 | - | - | - | - | - |
| CLMR [48] | 73.5 | 17.0 | 73.5 | 12.6 | 84.6 | 16.2 | 81.3 | 26.4 | - | - | - | - | - |
| Jukebox-5B [15; 57] | - | - | - | - | - | - | - | - | 5.1* | 4.9* | 4.1* | 2.7* | - |
| MULE [36] | 76.6 | 19.2 | 78.0 | 15.4 | **88.0** | **20.4** | 83.7 | 30.6 | - | - | - | - | - |
| HuBERT-base$^{music}$ [25] | 75.5 | 17.8 | 76.0 | 13.9 | 86.5 | 18.0 | 82.4 | 28.1 | 4.7 | 3.7 | 1.8 | 2.1 | 55.8 |
| data2vec-base$^{music}$ [2] | 76.1 | 19.2 | 76.7 | 14.3 | 87.1 | 18.8 | 83.0 | 29.2 | 5.5 | 5.5 | 4.1 | 3.0 | 59.9 |
| MERT-95M$^{K\text{-}means}$ | 77.2 | 19.6 | 75.9 | 13.7 | 87.0 | 18.6 | 82.8 | 29.4 | 5.6 | 5.6 | 4.0 | 3.0 | 62.9 |
| MERT-95M-public$^{K\text{-}means}$ | 77.5 | 19.6 | 76.2 | 13.3 | 87.2 | 18.8 | 83.0 | 28.9 | 5.5 | 5.5 | 3.7 | 3.0 | 63.0 |
| MERT-95M$^{RVQ\text{-}VAE}$ | 77.5 | 19.4 | 76.4 | 13.4 | 87.1 | 18.8 | 83.0 | 28.9 | 5.5 | 5.5 | 3.8 | 3.1 | 63.7 |
| MERT-330M$^{RVQ\text{-}VAE}$ | 78.1 | 19.8 | 76.5 | 14.0 | 86.7 | 18.6 | 83.4 | 29.9 | 5.3 | 5.6 | 3.6 | 3.0 | **64.7** |
| (Previous) SOTA | **78.8** | **20.2** [1] | **78.6** | **16.1** [36] | 87.7 | 20.3 [1] | **84.3** | **32.1** [36] | **9.3** | **10.8** | **10.4** | **6.4** [44] | 64.5 |

## 5 RESULTS ANALYSIS

### 5.1 PERFORMANCE & EFFICIENCY OF MERT MODELS

The results on all the downstream tasks are provided in Tab. 1 and Tab. 2. As suggested by the average scores in Tab. 2, MERT-330M$^{RVQ-VAE}$ achieves the same score as the combination of the previous SOTAs (from 10 different models even including supervised methods) and becomes the new SOTA on 4 metrics. It is also noteworthy that the other smaller MERT-95Ms still have comparable performance. Generally, MERTs perform well on tasks focusing on local-level musical information such as beat, pitch and local timbre such as singer information, and remain competitive on the other tasks requiring more global-level information, such as music tagging, key detection, and genre classification. This indicates the blending of acoustic and musical teachers could provide comprehensive guidance for the understanding of music recordings, though pre-trained in only a 5-second context length. Nevertheless, the performances of our models in tasks with more global music information are close to strong baselines, suggesting MERT models are capable of recognising global patterns well, thanks to the relative position embeddings and the contextualisation of the transformers.

In addition, our models demonstrate good results with limited data, even when training with public data that may lack enough diversity. MERT-95M-public and MERT-95M are both trained on a ~1k hour dataset and give competitive performance compared with the SOTA and MERT-330M, proving that MERT can converge effectively and learns generalisable music representations with

limited training data. Moreover, the `MERT-95M-public` is trained with Music4ALL (Santana et al., 2020), a 910-hours public music dataset with mainly pop music and lack of diversity in music style, and shows comparable performance to other settings. In particular, its performance does not have a significant difference besides genre classification on GTZAN compared to `MERT-95M`.

We evaluate the performance of the $MERT^{RVQ-VAE}$ model with a parameter size of 95M and 330M, given the use of the EnCodec feature enables us to scale up the dataset compared to the K-means. The results demonstrate that increasing the model size to 330M yields improved performance or maintains similar performance compared to $MERT-95M^{RVQ-VAE}$ (less than 0.1%) on most of the tasks besides beat tracking. More importantly, the lightweight sizes of MERTs open up new possibilities for transferring one general understanding model for large-scale classification or sequence labelling MIR tasks. MERT series models achieve better or comparable performance with only 1.9% (95M) and 6.6% (330M) parameters compared to the self-supervised baseline Jukebox-5B (Dhariwal et al., 2020). Even when our evaluation is in probing setting, most models could not be trained on sequence labelling tasks like beat tracking or source separation with affordable computational costs except for MERT and baseline models with similar architecture (Hsu et al., 2021; Baevski et al., 2022).

Table 3: Evaluation Results from Models Trained with Different Teacher Settings. Models labeled with $^{\triangle 2}$ and $^{\blacktriangle 2}$ suggest that the K-means teachers are trained with the features from $^{\triangle 1}$ and $^{\blacktriangle 1}$ models. All the listed models are sized in (95M) and not augmented with the in-batch noise mixture.

| Acoustic Teacher | Acoustic Target Class | Musical Teacher | MTT Tagging | | GS Key | GTZAN Genre | EMO Emotion | | Avg. |
|---|---|---|---|---|---|---|---|---|---|
| | | | ROC | AP | $Acc^{Refined}$ | Acc | $R2^V$ | $R2^A$ | |
| $K\text{-means}^{MFCC}$ | 100 | | 89.8 | 36.3 | 15.1 | 66.2 | 39.6 | 67 | 49.4 |
| $K\text{-means}^{MFCC}$ | 500 | | 90.3 | 38 | 17 | 70 | 40.6 | 67.5 | 51.3 |
| $K\text{-means}^{MFCC}$ | $2000^{\triangle 1}$ | | 90.2 | 37.6 | 15.6 | 70 | 44.3 | 67.6 | 51.4 |
| $K\text{-means}^{Logmel+Chroma}$ | $300 + 200$ $^{\blacktriangle 1}$ | N/A | 90.5 | 37.6 | 55.1 | 75.2 | 40.1 | 68.2 | 62.1 |
| $K\text{-means}^{MFCC}$ | $2000^{\triangle 2}$ | | 90.4 | 37.5 | 16.1 | 68.3 | 43.9 | 67.7 | 51.0 |
| $K\text{-means}^{Logmel+Chroma}$ | $500^{\blacktriangle 2}$ | | 90.4 | 37.7 | 49.2 | 72.8 | 46.5 | 66.9 | 60.7 |
| $K\text{-means}^{MFCC+CQT}$ | 300+200 | | 89.4 | 35.3 | 53.2 | 69.0 | 45.8 | 66.8 | 60.2 |
| $K\text{-means}^{Logmel+Chroma}$ | $300 + 200$ | CQT | 90.6 | 38.4 | 65.0 | **78.6** | 53.1 | 68.7 | **67.3** |
| | $1024 \times 8$ $^{all\ codebook}$ | N/A | **90.7** | **38.7** | 60.5 | 72.8 | **55.3** | 69.0 | 65.0 |
| | $1024 \times 8$ $^{all\ codebook}$ | | 90.5 | 38.4 | 63.2 | 77.2 | 53.2 | **72.3** | 66.9 |
| RVQ-VAE | $1024$ $^{codebook7}$ | CQT | 88.6 | 34.4 | 63.5 | 62.1 | 33.3 | 53.2 | 57.6 |
| | $1024$ $^{codebook0}$ | | 90 | 36.7 | 59.4 | 67.2 | 39.7 | 64.5 | 60.5 |
| | $1024 \times 8$ $^{random\ codebook}$ | | 90.6 | 38.1 | **66.8** | 73.8 | 48.1 | 68.6 | 65.8 |

## 5.2 THE EFFECTIVENESS OF ACOUSTIC & MUSICAL TEACHER

As demonstrated in Tab. 3, we explore optimal combinations and selections of the teacher models in the MERT paradigm with a subset of downstream tasks following Castellon et al. (2021), including auto-tagging, key detection, genre classification, and emotion recognition.

We reproduce the original HuBERT (Hsu et al., 2021) setting on music datasets with the acoustic teacher $K\text{-means}^{MFCC\triangle 1}$ and the teacher $K\text{-means}^{MFCC\triangle 2}$ trained on features produced by HuBERT model from the first stage, similar to DeepCluster (Caron et al., 2018). We observe that such models perform poorly on the key detection and emotion recognition tasks even we increase the dimension of the MFCC features from 100 to 2000. As the re-clustering K-means does not bring significant improvement in the second stage pre-training, we stick to the ordinary one stage pre-training to study the influence brought by the teachers with less computational cost.

Given that the key information is highly related to the pitch classes of the audio, we then introduce such inductive bias by providing the K-means acoustic teacher with additional Chroma or CQT features, denoted as $K\text{-means}^{Logmel+Chroma\blacktriangle 1}$ and $K\text{-means}^{MFCC+CQT}$. The additional pitch information indirectly brought by the Chroma and CQT features immediately endow the model a certain of level of key detection ability and raise the accuracy from 15.6 to 55.1 and 53.2 while keeping or increasing performances on other tasks. This confirms that the potentials of transformer models can be better excavated from more dimensions by introducing extra pseudo prediction targets in the MLM scheme. Following such an intuition, it could be further assumed that designing a proper multi-task learning pre-training paradigm can guide the model to produce more general representations for various music

understanding tasks. We thus propose leveraging the CQT explicitly as a musical teacher to introduce harmonic inductive bias during the pre-training. Compared to models trained with only the acoustic teacher $^{\text{MFCC}\Delta\ 1}$ or K-means$^{\text{Logmel+Chroma}\blacktriangle 1}$, MERT models trained with the newly proposed CQT musical teacher that are naturally more aligned to music audio can achieve significant performance gains on not only the key detection task but also the tasks requiring the high-level information like genre classification and emotion recognition.

However, given that K-means models are difficult to scale up on large datasets due to memory and computational requirements, we use the RVQ-VAE model EnCodec (Défossez et al., 2022) as the final version of acoustic teacher without searching for the immeasurable hyper-parameter $K$. The EnCodec could intuitively provide more comprehensive acoustic information since the audio can be greatly recovered from the intermediate discrete codecs from the encoder by a neural decoder. We observe that leveraging only one top ($1024^{\text{codebook7}}$) or bottom layer ($1024^{\text{codebook0}}$) of the residual codebooks in RVQ-VAE already provide substantial information in pre-training, and the use of all layers in the codebooks allows the student models to learn richer acoustic patterns. Although the strategy of randomly accessing one of the codebooks for each batch can alleviate the use of GPU memory and lead to similar performance compared to using all of them at a time, the setting of predicting 8 coodebooks together is adopted for faster convergence in the finalised design. By replacing the acoustic teacher with RVQ-VAE, MERT achieves an average score of 66.9, similar to that of the K-means$^{\text{Logmel+Chroma}\blacktriangle 1}$ version (i.e., 67.3) while largely reducing the cost of scaling up K-means.

## 5.3 STUDY ON MUSICAL LOSS

Table 4: Evaluation Results for Musical Loss Study. The listed models are not augmented with the in-batch noise mixture.

| Parameter Size | Acoustic Teacher Model | Acoustic Target Class | Musical Loss Weight | MTT Tagging | | GS Key | GTZAN Genre | EMO Emotion | | Avg. |
|---|---|---|---|---|---|---|---|---|---|---|
| | | | | ROC | AP | Acc$^{\text{Refined}}$ | Acc | R2$^{\text{V}}$ | R2$^{\text{A}}$ | |
| 95M | K-means$^{\text{Logmel+Chroma}}$ | 300 + 200 | N/A | 90.5 | 37.6 | 55.1 | 75.2 | 40.1 | 68.2 | 62.1 |
| | | | 1 | 90.6 | 38.4 | 65.0 | **78.6** | 53.1 | 68.7 | **67.3** |
| | | | 2 | 90.6 | 38.1 | 62.7 | 66.9 | 45.5 | 67.9 | 62.7 |
| | | | 5 | 90.4 | 37.3 | **65.3** | 70.3 | 45.7 | 68.3 | 64.1 |
| 95M | RVQ-VAE | $1024\times 8$ $^{\text{all codebook}}$ | N/A | **90.7** | **38.7** | 60.5 | 72.8 | **55.3** | 69.0 | 65.0 |
| | | $1024\times 8$ $^{\text{all codebook}}$ | 1 | 90.5 | 38.4 | 63.2 | 77.2 | 53.2 | **72.3** | 66.9 |

We conducted a hyperparameter search to determine the optimal weight for the musical loss applied to masked audios in the k-means setting. In Table 4, we present the results of the varying musical loss weights, which uses the same evaluation setting in § 5.2. By adjusting the weight, we find that a weight of 1 yielded the best overall performance for the base model. We observe that when switching the acoustic teacher to RVA-VAE, the models performs slightly worse on GS than those with K-means. Overall, our study provides valuable insights into the impact of musical loss and different acoustic models on the performance of the acoustic language model. These findings can inform the future development of more effective and efficient models in the domain of acoustic processing.

## 6 CONCLUSION

In conclusion, our work underscores the potential of SSL for modelling raw music audio and the efficacy of our approach, MERT, in pre-training sizeable models. We present a novel paradigm that integrates RVQ-VAE and CQT teacher models, providing a unique blend of acoustic and musical information necessary for MLM-based pre-training for music understanding. This integration, bolstered by the application of an in-batch noise mixup data augmentation and Pre-LN, enables the learning of robust music representations with further training stability. The performance of the MERT model surpasses previous SSL baselines, achieving SOTA or comparable results across a wide range of MIR tasks while using significantly smaller parameter size. We anticipate that our method and the forthcoming public release of our codes and models will catalyse further research into the application of SSL in music audio, thereby broadening the scope and depth of human understanding of music. Despite being capable of handling longer sequences with relative positional embedding, our models are limited by the short 5-second training context, so our approach could be further improved for tasks requiring understanding extended musical contexts if trained on longer sequences.

## LIMITATION AND FUTURE WORK

Our models are trained using only 5-second audio signals due to constraints in computational resources and the extended length of audio signals. Despite these models being capable of handling longer sequences thanks to relative positional embedding, this approach could potentially limit their performance in tasks requiring a comprehensive understanding of extended musical contexts. We plan to continue training our models on a longer context once gaining access to more computing resources. Moreover, although we propose several techniques to improve the training stability for the acoustic pre-training, we still suffer from the gradient exploding issues with the half-precision training for settings with larger batch sizes and model sizes. In addition, we observe inverse-scaling effect in specific tasks while scaling-up to 330M, which indicates that our design could be further improved by stabilising the training.

## ACKNOWLEDGEMENT

This paper is a tribute to our talented friend Anqiao Yang, for his friendship and valuable advice to this work. Yizhi Li is a Ph.D. student fully funded by the Department of Computer Science, University of Manchester, UK. This work is partially funded by Theme based Research Scheme (T45-205/21-N), Research Grants Council of Hong Kong. Yinghao Ma is a research student at the UKRI Centre for Doctoral Training in Artificial Intelligence and Music, supported by UK Research and Innovation [grant number EP/S022694/1]. Emmanouil Benetos is supported by a RAEng/Leverhulme Trust Research Fellowship [grant number LTRF2223-19-106]. We acknowledge IT Services at The University of Sheffield for the provision of services for High Performance Computing.

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

# APPENDIX A    METHODOLOGY

## A.1    ROBUST REPRESENTATION LEARNING

**Algorithm 1** Pseudocode description of the pre-training loss calculation in Python style.

```
1  def loss_cal(x_batch, x_acoustic_labels):
2      # retrieve embeddings for acoustic class
3      y_VQ = embedding(x_acoustic_labels)
4      # prepare CQT targets
5      y_CQT = compute_CQT(x_batch)
6      # conduct in-batch mixture
7      x_noised = mixture(x_batch)
8      # compute the representations
9      z = MERT(x_noised)
10
11     # loss calculation
12     loss_acoustic = Cross_Entropy(z[mask_idx], y_VQ[mask_idx])
13     loss_musical = Mean_Square_Error(z[mask_idx], y_CQT[mask_idx])
14     return loss_acoustic, loss_musical
```

We introduce "in-batch noise mixup" for music SSL. The mixup augmentation refers to the audio clip being added up with a certain ratio of shorter audio excerpts to form an augmented single sample during pre-training, instead of using the original audio. We randomly sample the audio segments from the same batch and add them to audio at random positions according to some probability. Theoretically, sampling from the whole training dataset would provide more randomness and thus be more beneficial to the representation robustness, but we narrow the sampling pool to the same audio batch considering the limited computational resources. The mixup could enable the learning of more robust musical representations and force the model to focus on the useful musical source and to ignore the noise. A pseudocode implementation can be found in Algo. 1.

Using the same evaluation setting in § 5.2, we alter the in-batch mixup probability to evaluate whether it is affecting the performance of the model when compound with musical loss. We found the mixup probability provides worse results in $\texttt{MERT}^{\texttt{K-means}}$ but provides better performance for $\texttt{MERT}^{\texttt{RVQ-VAE}}$. Therefore, we determined a probability of $0.5$ to be suitable based on the average performance score, and set it as a hyper-parameter in the final released model. Such a phenomenon deserves more attention.

Table 5: Evaluation Results for Pre-training Setting Ablation Study.

| Parameter Size | Acoustic Teacher Model | Acoustic Target Class | Musical Loss Weight | In-batch Mixup Probability | MTT Tagging | | GS Key | GTZAN Genre | EMO Emotion | | Avg. |
|---|---|---|---|---|---|---|---|---|---|---|---|
| | | | | | ROC | AP | Acc$^{\text{Refined}}$ | Acc | R2$^{\text{V}}$ | R2$^{\text{A}}$ | |
| 95M | K-means$^{\text{Logmel+Chroma}}$ | 300 + 200 | N/A | N/A | 90.5 | 37.6 | 55.1 | 75.2 | 40.1 | 68.2 | 62.1 |
| | | | 1 | N/A | 90.6 | 38.4 | 65.0 | **78.6** | 53.1 | 68.7 | 67.3 |
| | | | 2 | N/A | 90.6 | 38.1 | 62.7 | 66.9 | 45.5 | 67.9 | 62.7 |
| | | | 5 | N/A | 90.4 | 37.3 | 65.3 | 70.3 | 45.7 | 68.3 | 64.1 |
| | | | 1 | 0.25 | 90.6 | 37.9 | **65.5** | 70.0 | 49.6 | 72.5 | 65.2 |
| | | | 1 | 0.5 | 90.7 | 38.6 | 64.9 | 72.8 | 45.3 | 71.9 | 65.2 |
| 95M | | 1024×8 $^{\text{all codebook}}$ | 1 | N/A | 90.5 | 38.4 | 63.2 | 77.2 | 53.2 | 72.3 | 66.9 |
| 95M | RVQ-VAE | 1024×8 $^{\text{all codebook}}$ | 1 | 0.5 | **91.0** | **39.3** | 63.3 | **78.6** | **60.0** | **76.4** | **68.8** |

## A.2 THE EFFECTIVENESS OF VANILLA RVQ-VAE RREPRESENTATION

Table 6: Evaluating the EnCodec RVQ-VAE Embeddings.

| Dataset | MTT | | GS | GTZAN | EMO | | VocalSet | VocalSet |
| Task | Tagging | | Key | Genre | Emotion | | Tech | Singer |
| Metrics | ROC | AP | Acc$^{\text{Refined}}$ | Acc | R2$^{\text{V}}$ | R2$^{\text{A}}$ | Acc | Acc |
| --- | --- | --- | --- | --- | --- | --- | --- | --- |
| RVQ-VAE Embedding | 83.4 | 26.2 | 12.1 | 36.5 | 10.3 | 47.4 | 46.3 | 69.4 |
| MERT-95M$^{\text{K-Means}}$ | 90.6 | 38.4 | 65.0 | 78.6 | 52.9 | 69.9 | 74.6 | 77.2 |
| MERT-95M$^{\text{RVQ-VAE}}$ | 91.0 | 39.3 | 63.5 | 78.6 | 60.0 | 76.4 | 74.2 | 83.7 |

To verify that the benefits brought by the MERT pre-training, we further evaluate the performances of the continuous representation from the encoder of the RVQ-VAE model, as shown in Tab. 6. These supplementary results indicate that the vanilla continuous representations alone are insufficient for a robust music understanding baseline.

## APPENDIX B   EXPERIMENT DETAILS

### B.1   DOWNSTREAM TASKS

We evaluate the models on 14 downstream tasks to provide a comprehensive view of our method and the comparison between baselines. The full descriptions of the datasets and tasks are given as follows.

**Music Tagging** involves determining which of a set of fixed tags apply to a particular song. Tag categories may include genre, instrumentation, mood, tempo (e.g. fast) or other tags. We used two large datasets: MagnaTagATune (MTT) (Law et al., 2009) and MTG-Jamendo (Bogdanov et al., 2019). For both datasets, we limit the tag vocabulary according to official instructions. We use all clips in MTT and MTG-Jamendo. Since many of the audio recordings among 5.5k MTG-Jamendo excerpts are longer than the 30s, we averaged the multiple embeddings computed with a sliding window as the overall embedding. The window length is set to the same default length as in every system. For MERT series, the window length is typically set to 30s. The metrics are the macro-average of ROC-AUCs and the average precision (AP) / PR-AUC among all top-50 tags.

**Key detection** predicts the tonal scale and dominant pitch level of a song. We use Giantsteps (Knees et al., 2015) as test set and a commonly-used subset of Giantsteps-MTG-keys dataset (Korzeniowski and Widmer, 2017) as the training and validation set. The splitting is the same as in (Castellon et al., 2021). The metric is a refined accuracy with error tolerance, giving partial credit to reasonable errors (Raffel et al., 2014).

**Genre classification** estimates the most appropriate genre for each given song. We report the accuracy of the GTZAN (Tzanetakis and Cook, 2002) dataset along with ROC and AP on MTG-Genre, since the former task is a multi-class classification and the latter is multi-label. We used the standard "fail-filtered" split (Kereliuk et al., 2015) for GTZAN.

**Emotion score regression.** The Emomusic dataset (Soleymani et al., 2013) contains 744 music clips of 45 seconds in length, each reported on a two-dimensional valence-arousal plane after listening, where valence indicates positive and negative emotional responses, and arousal indicates emotional intensity. We use the same dataset split as (Castellon et al., 2021). The official evaluation metric is the determination coefficient ($r^2$) between the model regression results and human annotations of arousal (EmoA) and valence (EmoV) (Soleymani et al., 2013). For inference, we split the 45-second clip into a 5-second sliding window and averaged the prediction.

**Instrument classification** is the process of identifying which instruments are included in a given sound. We use the Nsynth (Engel et al., 2017) and MTG-instrument datasets. The former is a monophonic note-level multi-class task with 306k audio samples in 11 instrument classes with accuracy as an indicator. The latter is a subset of MTG-Jamendo, containing 25k polyphonic audio tracks and 41 instrument tags; each track can contain multiple instruments and is evaluated on ROC and AP.

**Pitch classification** estimates which of the 128 pitch categories the given audio segment belongs to. We use the NSynth dataset for this task. Given these segments are short monophonic audio, this task is multi-class, and the accuracy is used as an evaluation metric.

**Vocal technique detection** involves identifying what singing techniques are contained in a given audio clip. We use the VocalSet dataset (Wilkins et al., 2018), which is the only publicly available dataset for the study of singing techniques. The dataset contains the vocals of 20 different professional singers (9 female and 11 male) who perform 17 different singing techniques in various contexts for a total of 10.1 hours. As the audio clips are divided into 3 seconds, the task only requires a judgement on the type of technique and not on the start and end of the technique. We used the same 10 different singing techniques as in Yamamoto et al. (2022) as a subset and used the same 15 singers as the training and validation sets and 5 singers as the test set. Since there is no accepted division between training and validation sets, we selected 9 singers as the training set and 6 singers as the validation set. All the 3-second segments that originate from the same recording are allocated to the same part of the split (e.g. all are in the training set). The evaluation metric is accuracy.

**Singer identification** identifies the vocal performer from a given recording. We use the VocalSet dataset for this task. We randomly divided the dataset into a training set, validation set and testing set based on a ratio of 12:8:5, all containing the same 20 singers.

**Beat tracking** is the process of determining whether there is a beat in each frame of a given piece of music. We use an offline approach to the binary classification, i.e. the model can use information following each frame to help with inference. The model needs to output frame-by-frame predictions at a certain frequency and post-process them using a dynamic Bayesian network (DBN) (Böck et al., 2016b) to obtain the final result. The DBN is implemented using madmom (Böck et al., 2016a). The dataset we use is GTZAN Rhythm (Marchand and Peeters, 2015). We also label the two adjacent frames of each label as beat, which is a common way of label smoothing in beat tracking to improve the performance of the model and to compare the SSL model fairly with the spin model. The model is evaluated using the f_measure implemented in mir_eval (Raffel et al., 2014), and the prediction is considered correct if the difference between the predicted event and the ground truth does not exceed 20ms. In this task, some models were trained on other datasets, and the full GTZAN set was used as the test set.

**Source separation.** Source separation aims to demix the music recording into its constituent parts, *e.g.*, vocals, drums, bass, and others. We adopt MUSDB18 (Rafii et al., 2017), a widely used benchmark dataset in music source separation. MUSDB18 contains 150 full-length music tracks ($\tilde{1}0$ hours), along with multiple isolated stems. We use 86 tracks for training, 14 tracks for validation, and 50 tracks for evaluation following the official setting in MUSDB18. During training, we randomly sample 6-second segments and apply random track mixing for augmentation. Due to the difficulty of this task, we adopt the baseline architecture in the Music Demixing Challenge (MDX) 2021 (Mitsufuji et al., 2022), which consists of three linear layers and three bi-directional LSTM layers. We directly compute the l2-loss between predicted and ground-truth spectrograms for optimisation. The metric for this task is the Source-to-Distortion Ratio (SDR) defined by MDX 2021 (Mitsufuji et al., 2022), which is the mean across the SDR scores of all songs.

## B.2 TESTING PROTOCOL DETAILS

Our aim is to explore the generality and standardisation of the framework. We, therefore, freeze the parameters of the pre-trained model to extract pre-trained features as fixed depth embeddings that are fed to each downstream task-specific prediction head. This allows for as lightweight a solution as possible for all tasks, thus testing whether the representations are easily reusable across different downstream tasks. In the following, we first describe the selected pre-trained baseline model, followed by the downstream model and training strategy.

In order to detect representations with relevant information about the downstream MIR task, we use these representations as input features to train a shallow supervised model on each task. For most tasks we use an MLP with one hidden layer, and for source separation, we use the baseline of the demixing data challenge described above, with the 3-layer LSTM used as post-processing. Since some representations may require different hyperparameter configurations to be successfully trained, we performed the following hyperparameter search for each mentioned SSL mainly based on MARBLE[2] benchmark, using the validation set for each downstream task.

- Model: {one-layer MLP with 512 hidden units, 3-layer LSTM (source separation only)}
- Batch size: {64}
- Learning rate: {1e-4, 5e-4, 1e-3, 5e-3, 1e-2}
- Dropout probability: {0.25}
- Optimizer: Default Adam optimizer
- Early Stopping: Fixed across all models with task-specific patience
- LR Scheduler: Reduce LR On Plateau, fixed across all models with task-specific patience

In addition, although we use the same hyperparameter grid for all tasks, the learning objectives vary from task to task. For the same task with a uniform dataset, if there are different evaluation metrics,

---

[2]https://marble-bm.shef.ac.uk

we will average the two evaluation metrics. We keep the best validation set results, and use the test set results as the final results of the benchmark.

## B.3 TRAINING INSTABILITY

In the experiments of scaling up to `MERT-330M` under mix precision training (fp16), we have explored several settings and plot the gradient norm, scale of loss, the MLM loss on acoustic targets, and the MLM loss on musical targets (see Fig. 2).

We first adopt the Pre-LN setting as in the HuBERT (Hsu et al., 2021) x-large model for stable training. However, the training crashed around 50K step under this vanilla solution from the speech model and thus we restart the pre-training at 40K step with gradient clipping threshold reduced from 10.0 to 1.0. The second run of Pre-LN lasted for 40K steps and crashed due to the same reason of reaching minimum loss scale.

We suspect the instability could be brought by the increased depth of the Transformer module. Following the strategies in DeepNorm (Wang et al., 2022a), we tried to alleviate the instability by initialising the Transformer with smaller values and enhancing the residual connection in the Post-LN. Unfortunately, such modification causes model collapse around 20K steps.

We then turned back to the stable Pre-LN setting and leveraged the attention relaxation trick proposed in Chen et al. (2021b). The additional scale constant in softmax calculation in the attention module alleviates the overflow problem and allows the final version of `MERT-330M` model to be trained stably over 100K steps.

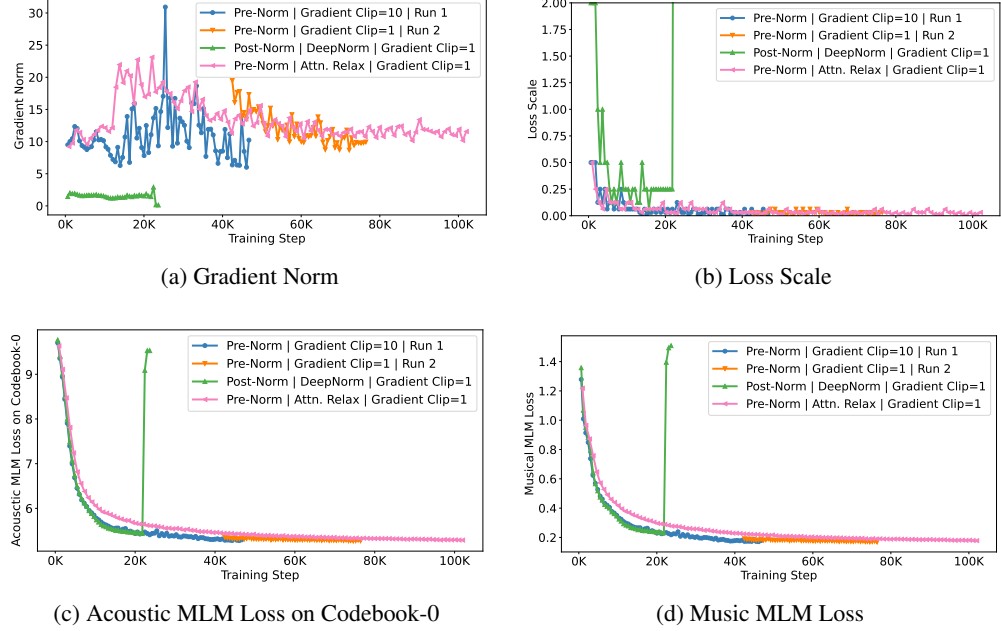

(a) Gradient Norm

(b) Loss Scale

(c) Acoustic MLM Loss on Codebook-0

(d) Music MLM Loss

Figure 2: Illustration of the Training Curves of Trials on Large (330M) Models. Only the acoustic MLM loss on codebook 0 in the RVQ-VAE is shown as the other seven show similar trends.

## APPENDIX C    REPRESENTATION VISUALISATION

We select two of our checkpoints, MERT-95M-publicK-means and MERT-330MRVQ-VAE, and visualise the GTZAN representations with genre annotation shown in Fig. 3, Fig. 4, Fig. 5 and Fig. 6. The top 6 and top 8 transformer output layers are used in the visualisation for MERT-95M-publicK-means and MERT-330MRVQ-VAE, correspondingly. The dimension reduction is achieved by the Uniform Manifold Approximation and Projection (UMAP)[3], whereas the representations from the training set are used to learn the dimension reduction mapping. We observe that representations from both of the checkpoints present a pattern of clustering according to the genre information under different layer settings. Interestingly, the representations from the higher layers do not necessarily show stronger genre-based clustering tendency, which suggests that 1) genre may not be the most abstractive labels for these music examples or 2) the top transformer layers focus more on the MLM pre-training objectives.

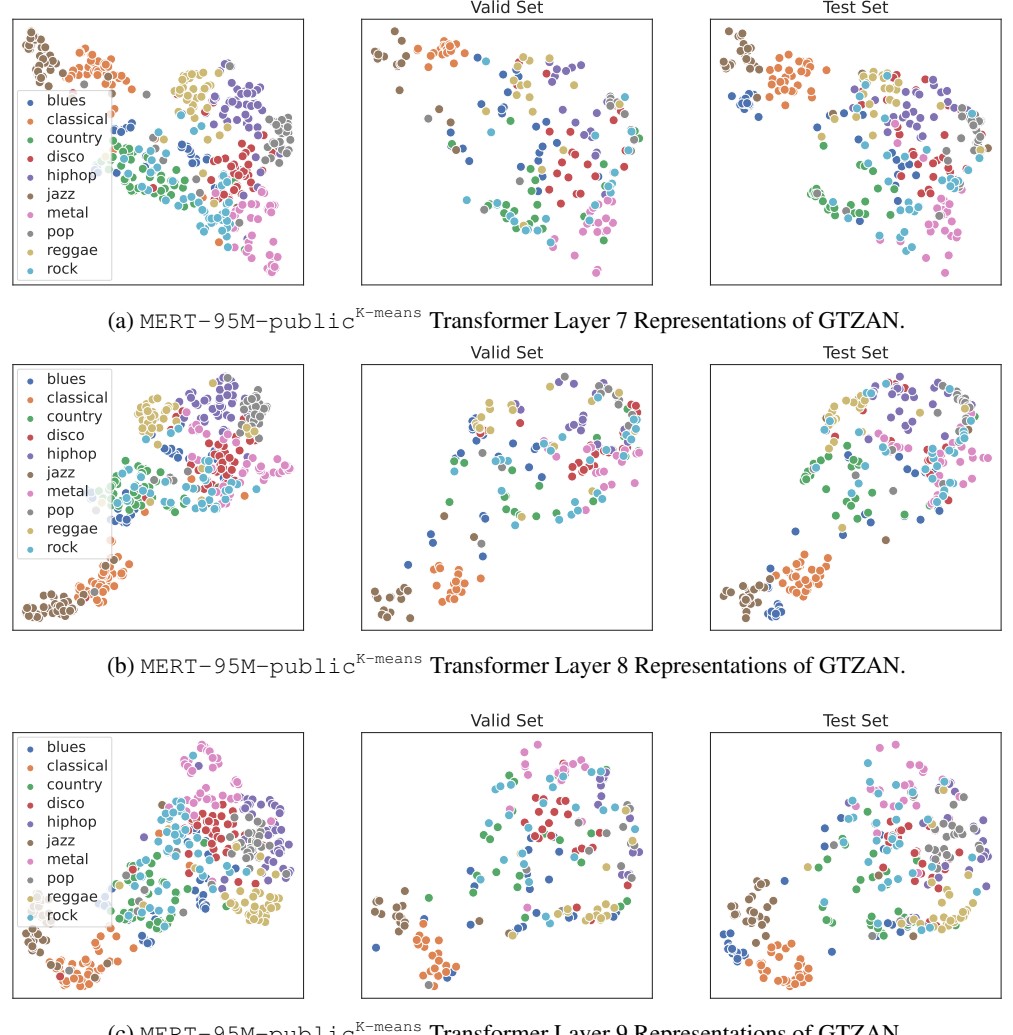

(a) MERT-95M-publicK-means Transformer Layer 7 Representations of GTZAN.

(b) MERT-95M-publicK-means Transformer Layer 8 Representations of GTZAN.

(c) MERT-95M-publicK-means Transformer Layer 9 Representations of GTZAN.

Figure 3: Illustration of the MERT-95M-publicK-means Layer 7 to 9 Pre-trained Representations.

---

[3]https://github.com/lmcinnes/umap

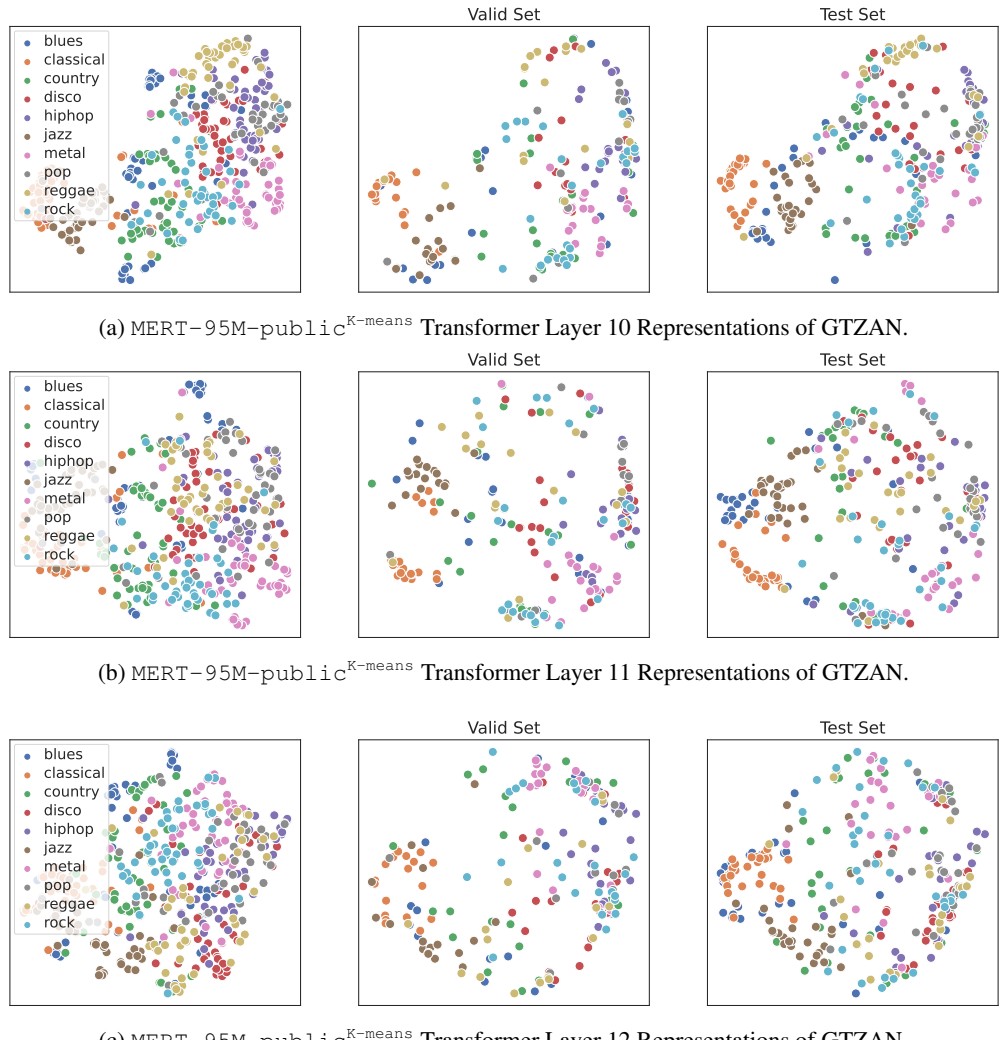

(a) MERT-95M-publicK-means Transformer Layer 10 Representations of GTZAN.

(b) MERT-95M-publicK-means Transformer Layer 11 Representations of GTZAN.

(c) MERT-95M-publicK-means Transformer Layer 12 Representations of GTZAN.

Figure 4: Illustration of the MERT-95M-publicK-means Layer 10 to 12 Pre-trained Representations.

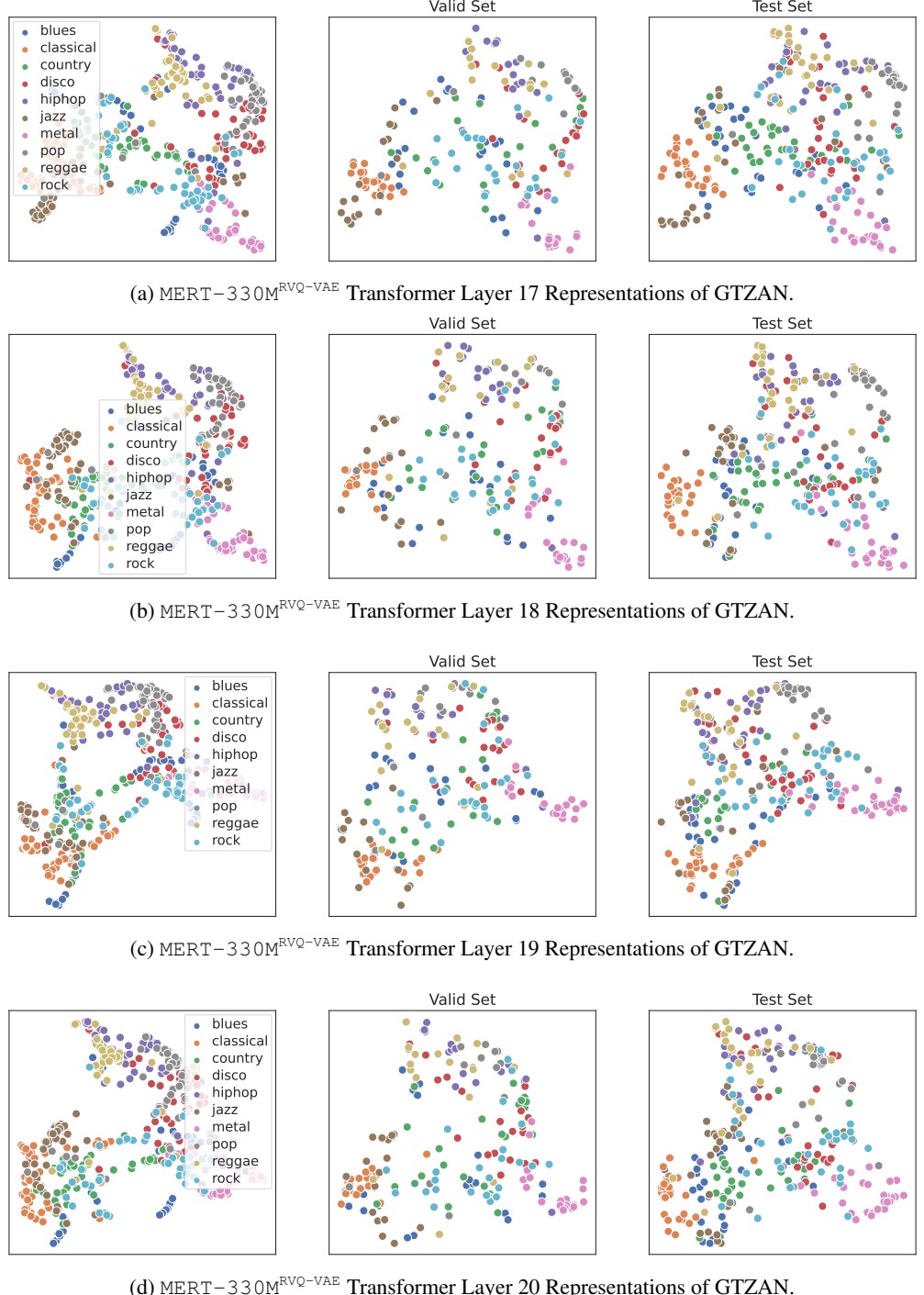

(a) MERT-330M$^{\text{RVQ-VAE}}$ Transformer Layer 17 Representations of GTZAN.

(b) MERT-330M$^{\text{RVQ-VAE}}$ Transformer Layer 18 Representations of GTZAN.

(c) MERT-330M$^{\text{RVQ-VAE}}$ Transformer Layer 19 Representations of GTZAN.

(d) MERT-330M$^{\text{RVQ-VAE}}$ Transformer Layer 20 Representations of GTZAN.

Figure 5: Illustration of the MERT-330M$^{\text{RVQ-VAE}}$ Layer 17 to 20 Pre-trained Representations.

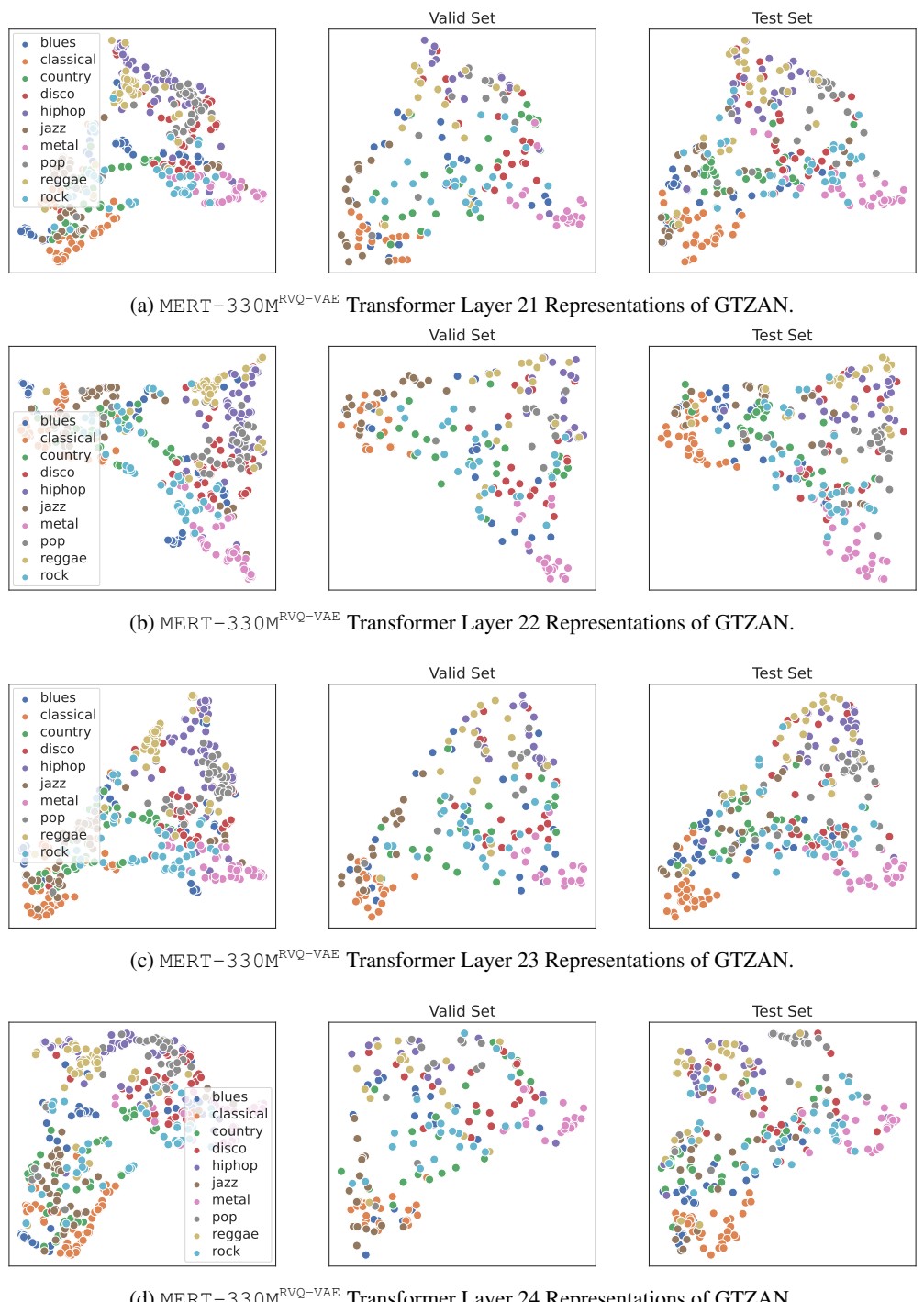

(a) MERT-330MRVQ-VAE Transformer Layer 21 Representations of GTZAN.

(b) MERT-330MRVQ-VAE Transformer Layer 22 Representations of GTZAN.

(c) MERT-330MRVQ-VAE Transformer Layer 23 Representations of GTZAN.

(d) MERT-330MRVQ-VAE Transformer Layer 24 Representations of GTZAN.

Figure 6: Illustration of the MERT-330MRVQ-VAE Layer 21 to 24 Pre-trained Representations.

## APPENDIX D   ETHICS

We have taken great care to ensure that our research adheres to ethical principles and guidelines in the codes of conduct. Specifically, we have not used inappropriate user information in the experiments. The audios are collected from open-access streaming services and open-source datasets, where the quality of the audios vary among 16KHz, 24KHz, and 48KHz. The audios would not be hosted and distributed. We believe that our work has the potential to contribute to positive social and scientific outcomes regarding the research of automatic music understanding.

