# OpenReview forum: "MERT: Acoustic Music Understanding Model with Large-Scale Self-supervised Training"
_ICLR.cc/2024/Conference — ICLR 2024 poster_

### Official Review · Reviewer_j7HH · 2023-10-30

**Soundness:** 2 fair
**Presentation:** 3 good
**Contribution:** 2 fair
**Rating:** 6
**Confidence:** 4

**Summary:**

The paper presents MERT - large scale self-supervised models for music tasks. The approach is built primarily on top of the speech SSL approach (Hubert). Aside from K-means, MERT proposes to use Encodec to get targets for SSL pre-training. Reconstruction loss on Constant-Q transform is also used for training. The pre-training is done on 1K hours of music data and then a 160K hour of music data. In terms of model size MERT offers two flavors: a 95M parameter model and a 330M parameter model. The MERT model is evaluated on a wide range of music tasks - Music tagging, music genre classification, instrument classification and so on.

**Strengths:**

– The paper presents self-supervised learned models for music audio. Better large scale SSL models for music are definitely desirable and the paper is a fair attempt at building such models. Moreover, the paper aims to make the model open-source which can definitely help in future development of SSL models for music.

– The paper describes the approach clearly. The datasets, downstreams tasks etc. are also well described.

– Discussions on challenges in scalability and training stabilities are also discussed. I think that’s a good topic to touch upon.

– The paper also does downstream evaluation on a variety of music datasets. This creates a really good benchmark for evaluations.

----
Increased score after rebuttal.

**Weaknesses:**

– The presented approach is primarily minor modifications of existing SSL models and the significance of the MERT training approach itself  is limited. It’s not fully established that the modifications over Hubert are really adding substantive improvements in performance.

–  “Computationally affordable”, “cost-effective”, “lightweight sizes”  etc. are frequently used for the proposed MERT but it is not really clear how all of these are attained for MERT. How is MERT more computationally efficient or lightweight  than say Hubert-Base. Aren’t the models similar ? What efficiencies are we expecting here? Is it just about K-Means vs uses of codebook from Encodec ? This aspect has been highlighted several times in the paper so it would be good properly establish (quantitatively ??) how MERT is better than others.

— Is it necessary to use all 8 layers of codebooks ? perhaps some additional experiments to better show how results vary with codebooks from different layers would be good.

— Related to the previous point, how about using Encodec itself as SSL representation. Codec is used in MERT as a teacher - can codec itself be used as a SSL model for music. Isn’t that a baseline one can have ?

— The “AVG” column in Table 2. What is it avg of ? of all the other columns ? I am not sure looking at avg of different types of metrics over different datasets is a good way to look at overall results.

– Comparing “HuBERT base” and MERT-95M^{K-Means} it seems that they are pretty similar.

– For all of the downstream experiments, is the full training set for each dataset used in the experiments ? I think some experiments on “limited training data” would be useful. Otherwise, these models are not really outperforming the supervised baselines – which does not full justify the SSL pre-training.

– In Sec 5.1, the paper describes that the model is doing better on local-level music tasks compared to global level tasks. Some more discussion and perhaps illustration of why this is happening would be super helpful.

**Questions:**

Please respond to the points in the weakness section.

---

> ### Author Response · Authors · 2023-11-18
>
> We are grateful for your thorough review and insightful suggestions.
>
> To address your Concerns:
> 1. **Novelty of Our Method Compared to HuBERT**: We understand your concern regarding the apparent similarity of our mask language modelling approach to HuBERT. However, we believe MERT significantly contributes to the field as it is the first model to apply masked language modelling specifically for music understanding. Our approach incorporates RVQ-VAE discrete labels for general semantic information and CQT reconstruction loss for pitch learning, tailored to the unique requirements of music modelling. The exploration of different combinations of acoustic features and cluster numbers for K-Means (as detailed in Table 3) is particularly noteworthy. While there are similarities with HuBERT in MLM tasks, MERT's distinct training logic and additional features, such as CQT reconstruction targets and transformer architecture optimizations for training stability (Section 4.3, paragraph 2), substantiate its novelty.
> 2. **Claim of Being Lightweight**: We apologise for any ambiguity in our earlier statement. Our comparison was intended against the performance of probing Jukebox5B (Castellon et al., 2021, Dhariwal et al., 2020), not HuBERT. We highlight MERT's efficiency by noting its significantly fewer parameters (1.9% and 6.6% compared to Jukebox5B) in Section 5.1, Paragraph 3.
> 3. **Necessity of Using All Codebooks**: Your question regarding the use of all eight layers of codecs in RVQ-VAE is well-founded. Specifically, we have tested:
>
>         a. Predicting the most informative codebook at layer 0=> score 60.5
>
>         b. Predicting the most informative codebook at layer 7 => score 57.6
>
>         c. Predicting one random layer of the codebook each time => score 65.8
>
>         d. Predicting all layers of codebook in parallel => score 66.9
>
>     Our investigation (referred in Table 3) shows that utilising all layers concurrently is optimal. However, we speculate that selecting a random layer for prediction and extending training could potentially yield similar results. Detailed analysis is presented in the last paragraph of Section 5.2.
>
> 4. **Effectiveness of EnCodec Representation**: Thank you for this valuable suggestion. We have conducted additional experiments to address this, which are now included in the revised paper. As Reviewer-7KYT also raised this question, we invite you to refer to point 2 in our [reply to Reviewer-7KYT](https://openreview.net/forum?id=w3YZ9MSlBu&noteId=PhDl8AtHDl) to avoid redundancy in our rebuttal.
>
> 5. **Averaging Metrics**: While averaging may not always be the optimal approach, it is a common practice in evaluating pre-trained models across established benchmarks (JukeMIR, GLUE, OpenCompass). We agree that more robust aggregation metrics could be developed for a more nuanced evaluation of pre-trained models.
>
> 6. **Downstream Training Set Usage for Evaluation**: The diverse training set sizes (from hundreds to 300K) across the 14 downstream tasks demonstrate the robustness of our evaluation method and generalisability of the model. We are also conducting few-shot setting evaluations, to further address your concerns. The results are anticipated in the next 2-3 days.
>
> 7. **Performance Gap Between Local and Global Level Tasks**: As discussed in Section 5.1, MERT is trained with 5-second contexts to balance performance and computational resources. Longer contexts might benefit certain tasks, but this involves a trade-off with batch size. Our decision aligns with recent findings in the field (Ma et al. 2023, On the Effectiveness of Speech Self-supervised Learning for Music, ISMIR), which is described at *Training Settings* paragraph in section 4.3.
>
> We hope these responses adequately address your concerns and clarify our approach.

---

> > ### Comment · Reviewer_j7HH · 2023-11-21
> > **comment on rebuttal**
> >
> > Thanks for the detailed response, especially additional experiments. Several of the points raised in the reviews have been addressed in the rebuttal. I have raised the score to reflect the improvements. Stronger results (compared to the base models and supervised learning), more music specific innovative modeling (rather than the combinations of different things from speech literature) etc would have made the paper stronger for ICLR.  Nevertheless, the paper is well done and a good benchmarking effort on SSL for Music tasks.

---

> > > ### Author Response · Authors · 2023-11-22
> > >
> > > Thanks the reviewer for acknowledging our contribution.
> > >
> > > We finish the zero-shot testing to further evaluate the generalisability of the pre-trained models, following the zero-shot implementation in [CLIP (Radford et al. 2021)](https://github.com/openai/CLIP/blob/main/notebooks/Prompt_Engineering_for_ImageNet.ipynb).
> > >
> > > By randomly sampling **5 data points** from each class, we acquire the cluster centroids by averaging the representations. Then the classification predictions are calculated according to the cosine similarity between the test samples and the cluster centroids. We only conduct the evaluation on the multi-class classification downstream tasks for simplicity.
> > >
> > > Even the zero-shot evaluation method might not fully represent the generalisability of the models, it still provides some useful insights. 1) The contrastive learning paradigm (CLMR) contributes , even with small model and training dataset. 2) Supervised pre-trained models on handcrafted logmel features (MusicNN) can be still regarded as strong baselines. 3) Pre-training with large-scale datasets incorporates with music domain knowledge (MERT) benefit somehow fill the gap brought by parameter sizes, and show good generalisability.
> > >
> > > *Table R2*
> > > | Model              | GTZAN-Genre_{ACC} | Vocal-Tech_{ACC} | GS_{Refined ACC} |
> > > | ------------------ | ----------------- | ---------------- | ---------------- |
> > > | MusiCNN            | 59.1              | 29.3             | 9.0              |
> > > | CLMR               | 51.0              | 31.2             | 9.1              |
> > > | Jukebox-5B         | 48.7              | 35.6             | 13.7             |
> > > | MERT-95M^{RVQ-VAE} | 60.7              | 37.9             | 20.1             |
> > >
> > > We hope the results could be helpful to the reviewers and the readers to better understand the behaviour of the acoustic pre-trained models.

---

> ### Comment · Area_Chair_vo1a · 2023-11-20
>
> Hello, reviewer. Please review the author's response to see whether it addresses your concerns.

---

### Official Review · Reviewer_Md8h · 2023-11-01

**Soundness:** 3 good
**Presentation:** 3 good
**Contribution:** 2 fair
**Rating:** 8
**Confidence:** 3

**Summary:**

This paper describes self-supervised learning technique for music audio. The most relevant previous work is HuBERT in speech domain which uses masked prediction learning using codeword of audio features. The authors applied this method to music audio and developed several techniques on top of the reference model. Basically, the most contributions lie on how they build codewords specifically dedicated to music audio. To do that, they mainly compared three ways which are MFCC-based codeword (K-Means), LogMel+Chroma-based codeword (K-Means), and EnCodec. MFCC and Log-Mel-spectrogram mainly captures timbre information from audio, so they utilized Chroma to compensate tonal information of music. Also, there has been a previous work called EnCodec which is a pre-trained codec encoder designed for music audio, so that this model already has some ability to capture both timbre and tonal information. Also, they added CQT loss to further enhance pitch and chord level information. In the end, the authors verified the models on 14 Music Information Retrieval tasks (mostly segment-level tasks, not note-level or frame-level). The results showed that EnCodec-based approach was the best performing model.

**Strengths:**

The strengths of the paper comes from how the authors tailored the previously proposed method to music audio domain. To do that, they tested various music audio specific techniques such as Chroma, CQT, and EnCodec. The results showed that these additional method improved the model performance on several downstream tasks that are more related to pitch, chord, tonal information of music. For the tasks where timbre information is important, the effect of using these tonal features is marginal.

**Weaknesses:**

The weaknesses of the paper is on novelty. If we see the results in Table 1 and 2, the trends are quite predictable even though the proposed method achieved SOTA performance on 3 tasks out of 14 tasks. For the models that doesn't utilize chord information, still those models achieve good performance on tasks where timbre is important (such as tagging, genre, mood, theme), however, if any methods includes to use this kind of information, then it shows good performance on both timbral and pitch related tasks. Also, it seems many performance boosts are made through the EnCodec, I think the novelty of the approach itself is a bit weak.

**Questions:**

If the used split of each downstream tasks can be written more in detail in Appendix, it would be better.
In Section 4.1, where GTZAN and MTG-Genre downstream task's metric is explained, only ROC and AP is mentioned, I think accuracy can be added.
In Section 4.3, "1.5 and 5.5 hours" is not a batch.

---

> ### Author Response · Authors · 2023-11-20
>
> **Addressing the Concerns about Novelty**:
>
> We are grateful for your remarks on the novelty of our approach. While the outcomes in Tables 1 and 2 may seem predictable, the true innovation of our work lies in the adaptation and integration of music-specific techniques, such as CQT reconstruction loss and the exploration of optimal pseudo targets for Masked Language Modeling (MLM) tasks, into the acoustic pre-training. This approach goes beyond mere incremental additions; it signifies a substantial advancement in adapting existing methods to the complexities of music understanding, a field that has been less explored than general audio processing.
>
> Furthermore, music audios contain a rich tapestry of elements, demanding more extensive training data compared to speech models. We have delved into maintaining stable pre-training on large-scale datasets, with specific architectural modifications and analysis detailed in Section 4.3 and Appendix B.3. This exploration is a testament to our commitment to advancing music audio modelling.
>
> At first glance, the performance improvements in MERT might seem attributable solely to the EnCodec model. However, we contend that these gains stem primarily from our refined pre-training stage. This is evidenced by:
> - The superior performance of MERT^{K-Means} in non-pitch-oriented tasks, such as instrument detection (Tables 1 & 2), compared to HuBERT trained on similar music data. This indicates the efficacy of our pre-training stage enhancements.
> - Additional verification showing that using only continuous embeddings from EnCodec results in a significant performance gap compared to both MERT^{K-Means} and MERT^{RVQ-VAE} (cf. [Table R2 in the reply to Reviewer 7YKT](https://openreview.net/forum?id=w3YZ9MSlBu&noteId=PhDl8AtHDl)). This underscores that MERT's learning extends beyond just the pseudo labels and encompasses a broader spectrum of information during the self-supervised pre-training phase.
>
> **Responses to Questions**:
> 1. *Downstream Tasks Metrics*: In Section 4.1, we've outlined metrics for downstream tasks by type. We agree with the reviewer that including a wider range of metrics would enhance the comprehensiveness of our evaluation. Our metric selection aligns with prior research to facilitate reproducibility and fair comparison. Comprehensive details of the training, validation, and test splits are provided in Appendix B.1.
> 2. *Batch Size Terminology*: The use of “1.5 and 5.5 hours” to describe audio contained in a training batch follows precedents set by works like HuBERT (Hsu et al. 2021) and WavLM (Chen et al. 2022). This approach allows readers to easily compare the scale of training batches in terms of audio content, complemented by the provided GPU numbers and clip length information.
>
> **Closing Remarks**
>
> We are dedicated to contributing significantly to the field of music audio modelling. The amendments made in response to your insights are aimed at enriching the content and understanding of our research. We are hopeful that these enhancements will elevate our paper to meet the acceptance criteria.

---

> ### Comment · Area_Chair_vo1a · 2023-11-20
>
> Hello, reviewer. Please review the author's response. You are concerned about the novelty of the paper. Does the author's response address your concern?

---

> > ### Comment · Reviewer_Md8h · 2023-12-01
> >
> > Dear authors, thank you for detailed explanation on your efforts and distinction from the previous work. As your detailed explanation, I agree the novelty of the paper is enough especially on tailored pre-training method for music audio and also the effect of this method (rather than it mostly comes from encodec feature).

---

### Official Review · Reviewer_EaeP · 2023-11-05

**Soundness:** 3 good
**Presentation:** 4 excellent
**Contribution:** 3 good
**Rating:** 8
**Confidence:** 3

**Summary:**

This paper proposes a self-supervised model for acoustic music understanding based on similar self-supervised learning paradigms in speech processing. The authors provide extensive comparison on 10 different MIR tasks some of which require understanding the local characteristics (such as pitch and beat), whereas some require a track-level understanding (such as genre, or emotion).

The authors experiment with two different teacher paradigms. They work with one 95M parameter model that is trained on publicly available music, and another larger model with 330M parameters that they train on 160k hours of music mined from internet. They compare their variants against the state of the art, and show that the model achieves similar or better results compared to the current state-of-the-art.

**Strengths:**

- Extensive comparison between different models, conditions and # parameters on various MIR tasks.
- The results indicate the strengths of the proposed model (e.g. efficacy on tasks that require local-level musical information) as well as the limitations (e.g. 5 second excerpts)
- Provide a strong baseline for future research on self-supervised learning on acoustic music that is comparable to the current state of the art.
- Extensive literature review, which facilitates to convey the basis of the work as well as the motivations
- The authors explain issues they have faced while training the model, and also how they mitigated these issues, which is invaluable for future research. See the Training Stability part in Section 4.3. as an example.
- Open source code, experiments and dataset (where shareable)
- The language is appropriate for an engineering work, and the paper is easy to follow.

**Weaknesses:**

- Works on short excerpts. The authors argue that this limitation could be overcome in future work.

**Questions:**

- We mined 160K hours of music recordings from the Internet ...

What are the typical sources for mining? Youtube, streaming services, Freesound, or something else? What is the typical audio quality? Are they copyrighted, or not? Do you keep the audio or only the relevant features (MFCC, CQT?)

- Some references are not well formatted and/or they miss key information (in particular the conference). Examples:

Alonso-Jimenez, P., Serra, X., and Bogdanov, D. (2022).
Bogdanov, D., Won, M., Tovstogan, P., Porter, A., and Serra, X. (2019)
Chen, W., Keast, J., Moody, J., Moriarty, C., Villalobos, F., Winter, V., Zhang, X., Lyu, X., Freeman,
E., Wang, J., et al.

- While Table 1-2 are compact and informative, it's impossible to track the references apart from following the hyperlinks as the Reference formatting do not include the number.

- Although they should be known in general, I would suggest the authors to mention the full name of all the metrics such as R2 or ROC used in the experiments.

In addition, some of the "previous SOTA" (e.g. 26, 36) are still the best. Wouldn't it mean that they are still the state-of-the-art?

- Appendix D - Ethics. I think there should be a mention of music copyrights here, in particular the implications about mining music from the Internet.

Below are minor suggestions and nitpicks that I'd like to provide for the sake of completeness. They do not contribute to my decision on the paper.

- The writing switches between British and American spelling, e.g. "masked language modeling" vs. "masked language modelling."
- Nitpick: Page 4 "data sample in a speech or language dataset..." -> the dataset doesn't have to be speech or language, e.g. it can contain instrumental music.
- "Additionally, we present the advanced SOTA for each task including" -> This phrase could be read as the proposed model advances the state of the art for all tasks, which is not necessarily the case. If I understand correctly, "the current SOTA" is a better wording.
- "... longer contexts if required" ->  longer contexts, if required (missing comma)
- Page 17 is almost fully empty.
- Figures 2 - 6 are very useful, however, they are not suitable for color-blind readers. I would suggest to change the line/marker styles for each  element in the legends.

---

> ### Author Response · Authors · 2023-11-19
>
> We are grateful for your positive evaluation of our work and appreciate the opportunity to discuss aspects of our methodology further.
>
> **Regarding Short Training Contexts**: We acknowledge your point about the potential benefits of longer segments in pre-training. Our current use of a 5-second context is a starting point, and we agree that extending this to capture more global information in music audio would be advantageous. Recent advancements in efficient transformer architectures offer promising avenues for achieving this with improved memory efficiency. Additionally, we are considering alternatives to the Conv1D encoder, which currently poses challenges in terms of stability and memory consumption, to facilitate longer context windows.
>
>
> Responses to Review Questions:
>
> **Audio Collection and Quality**: We sourced audio from open-access streaming services and open-source datasets. The audio quality varies, with sampling rates of 16KHz, 24KHz, and 48KHz, necessitating upsampling or downsampling for different training settings. We ensured quality through random sampling and conducting hearing tests from the dataset. Related ethical considerations are added in Appendix D.
>
> **Reference Formatting**: Thank you for pointing out the issue with reference formatting. We have added the missing conference information and corrected the numbering in the bibliography.
>
> **Terminology Clarifications**: We have reintroduced the full names of ROC and R2 in the experimental section for clarity.
>
> **Describing “Previous SOTAs”**: We appreciate your observation on the use of "Previous SOTAs". To avoid potential confusion, we have revised Table 2 to read “(Previous) SOTA”.
>
> **General Formatting and Language Consistency**: Based on your suggestions, we have:
>
> - Unified the language to British English.
> - Modified the phrase on page 4 from “speech and language dataset” to “sequential dataset”.
> - Changed “advanced SOTA” to “current SOTA”.
> - Corrected punctuation and added missing commas.
> - Improved figure accessibility by adopting markers and a new colour palette suitable for colour-blind users.
>
> We are thankful for your detailed formatting suggestions, which have been incorporated into the latest version of our paper. We hope that the challenges and solutions we have shared will be valuable for future research in music audio modelling.

---

### Official Review · Reviewer_7KYT · 2023-11-09

**Soundness:** 4 excellent
**Presentation:** 4 excellent
**Contribution:** 3 good
**Rating:** 8
**Confidence:** 4

**Summary:**

This paper proposed MERT, an SSL  model for music representation. The model is based on MLM training such as Hubert, and utilizes multiple pseudo targets during the pretraining stage such as k-means, constant-Q Transformation, and neural codec codes. By combining different self-supervised targets, experiments on 14 diverse downstream MIR tasks show that MERT is able to extract good representation for MIR tasks and attains SOTA overall scores. The checkpoints and code are open-source.

**Strengths:**

1. The method is simple but effective. The paper demonstrates the importance of choosing appropriate pseudo targets.
2. Considers a diverse set of MIR tasks for evaluation, providing a good standard that can be followed by future works.
3. Unlike many previous closed-source works, this paper has made the checkpoints publicly available, which is a significant contribution to the research community.

Overall, the paper is well-written with a clear goal and provides sufficient experiment results to support the claims. Although the conclusions are not surprising, the work is still significant for the related research community from a practical perspective due to its reproducibility and accessibility.

**Weaknesses:**

There are 2 minor concerns.
1. Since RVQ-VAE is pre-trained on a larger dataset, comparing MERT with CQT/k-means and MERT with RVQ-VAE is somewhat unfair.
2. The experiment should verify how RVQ-VAE code performs on downstream tasks to prove that the proposed MLM training phase with RVQ-VAE code as the target is required. Otherwise, one can directly utilize codes and codebook vectors from RVQ-VAE as upstream representations instead of MERT.

**Questions:**

1. How do you calculate "Previous SOTA average score" in Table 2? The number did not match any baselines listed in the table, is it referenced from another work?

---

> ### Author Response · Authors · 2023-11-18
>
> We sincerely appreciate the reviewer's recognition of our method's effectiveness.
>
> For addressing the reviewer's concerns:
> 1. **On Potentially Unfair Comparison between Pre-Training Targets**: We acknowledge your observation regarding the comparison between RVQ-VAE and K-Means targets. While it may appear unfair at first glance due to the variance in training-data size, it's important to note that K-Means models benefit from handcrafted acoustic features, providing a distinct inductive bias. This aspect makes a direct comparison with self-supervised RVQ-VAE not entirely straightforward. However, we concur with your insight that RVQ-VAE models, given their unsupervised nature, potentially have higher capacity limits, especially in learning from large-scale datasets.
> 2. **Performance of Vanilla RVQ-VAE Representation**: We have conducted additional experiments to evaluate the audio embeddings from the RVQ-VAE encoder, which are the continuous vectors used for quantization to discrete codecs. These supplementary results, now included in Table R1, indicate that the codec continuous representations alone are insufficient for a robust music understanding baseline. We are grateful for your suggestion of this complementary experiment, which substantiates the effectiveness of the self-supervised learning phase. Please note, these findings and discussions have also been added to Appendix A.2 in the revised manuscript for the benefit of future readers.
>
> **Table R1: Evaluating the EnCodec RVQ-VAE Embeddings**
>
> | Model               | MTT_{ROC} | MTT_{AP} | GS_{ACC} | GTZAN-Genre_{ACC} | EMO_{R2V} | EMO_{R2A} | Vocal-Tech_{ACC} | Vocal-Singer_{ACC} |
> | ------------------- | --------- | -------- | -------- | ----------------- | --------- | --------- | ---------------- | ------------------ |
> | RVQ-VAE  Embedding  | 83.4      | 26.2     | 12.1     | 36.5              | 10.3      | 47.4      | 46.3             | 69.4               |
> | MERT-95M^{K-Means}  | 90.6      | 38.4     | 65.0     | 78.6              | 52.9      | 69.9      | 74.6             | 77.2               |
> | MERT-95M^{RVQ-VAE}  | 91.0      | 39.3     | 63.5     | 78.6              | 60.0      | 76.4      | 74.2             | 83.7               |
>
>
> 3. **Clarifications on Table 2 Presentation**: Regarding the average score calculation in Table 2, we chose this approach as all metrics positively correlate with task performance. We apologise for the missing bibliography numbers; this was an oversight, and we intended to use numbered hyperlinks for easier reading. This error has been rectified in the revised manuscript.
>
> We trust that these clarifications and additions adequately address your concerns and enhance the paper's contribution. Your feedback has been invaluable in improving our work, and we are grateful for your thorough and constructive review.

---

### Meta-Review · Area_Chair_vo1a · 2023-12-02

**Metareview:**

Summary of Scientific Claims and Findings: The paper presents MERT, an SSL model tailored for music representation. MERT is based on MLM training, similar to Hubert, and utilizes multiple pseudo targets during its pretraining stage, including k-means, Constant-Q Transformation, and RVQ-VAE codec codes. The main claim is that MERT, by integrating various self-supervised targets, extracts effective music representations.


Strengths of the Paper:
(1) Methodological Simplicity and Effectiveness: The paper's approach is straightforward yet demonstrates significant effectiveness in music representation.
(2) Comprehensive Evaluation: The paper includes a wide range of tasks for evaluation, setting a solid benchmark for future research.
(3) Open-Source Contribution: The authors have made their code and checkpoints publicly available, enhancing the paper's practical value to the research community.


Weaknesses of the Paper:
The incremental nature of MERT's modifications over existing SSL models like HuBERT raises questions about the paper's novelty.

**Justification For Why Not Higher Score:**

The paper demonstrates the effectiveness of the SSL learning phase and the importance of music-specific feature integration. However, the incremental nature of improvements over existing models like HuBERT and the reliance on the existing EnCodec model for significant performance boosts raise questions about the novelty of the approach.

**Justification For Why Not Lower Score:**

The paper introduces MERT, a novel SSL model for music representation, demonstrating strong performance on diverse tasks. The use of multiple pseudo targets during pre-training effectively improves the model's capabilities. The open-source availability of checkpoints and code enhances the paper's utility to the research community. Given these points, the paper's strengths outweigh its minor weaknesses (novelty), and it contributes positively to the field of music representation learning.

---

### Decision · Program_Chairs · 2024-01-16

Accept (poster)